# The Fluctuation of Process Gasses Especially of Carbon Monoxide during Aerobic Biostabilization of an Organic Fraction of Municipal Solid Waste under Different Technological Regimes

**Sylwia Stegenta-Dąbrowska [1,]*, Jakub Rogosz [1], Przemysław Bukowski [1], Marcin Dębowski [1], Peter F. Randerson [2], Jerzy Bieniek [1] and Andrzej Białowiec [1]**

[1] Faculty of Life Sciences and Technology, Wrocław University of Environmental and Life Sciences, 37a Chełmońskiego Str., 51-630 Wrocław, Poland; rogoszj@gmail.com (J.R.); przemyslaw.bukowski@upwr.edu.pl (P.B.); marcin.debowski@upwr.edu.pl (M.D.); jerzy.bieniek@upwr.edu.pl (J.B.); andrzej.bialowiec@upwr.edu.pl (A.B.)

[2] School of Biosciences, Cardiff University, Sir Martin Evans Building, Museum Avenue, Cardiff CF10 3AX, UK; randerson@cardiff.ac.uk

\* Correspondence: sylwia.stegenta@upwr.edu.pl; Tel.: +48-71-320-5974

**Abstract:** Carbon monoxide (CO) is an air pollutant commonly formed during natural and anthropogenic processes involving incomplete combustion. Much less is known about biological CO production during the decomposition of the organic fraction (OF), especially originating from municipal solid waste (MSW), e.g., during the aerobic biostabilization (AB) process. In this dataset, we summarized the temperature and the content of process gases (including rarely reported carbon monoxide, CO) generated inside full-scale AB of an organic fraction of municipal solid waste (OFMSW) reactor. The objective of the study was to present the data of the fluctuation of CO content as well as that of $O_2$, $CO_2$, and $CH_4$ in process gas within the waste pile, during the AB of the OFMSW. The OFMSW was aerobically biostabilized in six reactors, in which the technological regimes of AB were dependent on process duration (42–69 days), waste mass (391.02–702.38 Mg), the intensity of waste aeration (4.4–10.7 $m^3 \cdot Mg^{-1} \cdot h^{-1}$), reactor design (membrane-covered reactor or membrane-covered reactor with sidewalls) and thermal conditions in the reactor (20.2–77.0 °C). The variations in the degree of waste aeration ($O_2$ content), temperature, and fluctuation of CO, $CO_2$, and $CH_4$ content during the weekly measurement intervals were summarized. Despite a high $O_2$ content in all reactors and stable thermal conditions, the presence of CO in process gas was observed, which suggests that ensuring optimum conditions for the process is not sufficient for CO emissions to be mitigated. In the analyzed experiment, CO concentration was highly variable over the duration of the process, ranging from a few to over 1,500 ppm. The highest concentration of CO was observed between the second and fifth weeks of the test. The reactor B2 was the source of the highest CO production and average highest temperature. This study suggests that the highest CO productions occur at the highest temperature, which is why the authors believe that CO production has thermochemical foundations.

**Dataset:** The following are available online at www.mdpi.com/xxx/s1

**Dataset License:** CC-BY-NC

**Keywords:** aeration rate; aerobic biostabilization; carbon monoxide; organic fraction; municipal solid waste; GHG emission; semipermeable membranes

## 1. Summary

The content of biodegradable matter, in other words, the organic fraction (OF) in municipal solid waste (MSW) is a factor limiting the possibility of MSW landfilling. As a result, it is necessary to apply the processes of biological waste treatment prior to landfilling. Biological treatment of waste involves the use of microbiological metabolic processes to bring a decomposition or transformation of organic substances contained in the waste into products that can be either returned to the natural cycle or prepared for safe landfilling. One of the solutions is an aerobic biostabilization (AB) of the organic fraction of municipal solid waste (OFMSW). Implementation of the AB of OFMSW should be based on an optimization of the technological parameters and monitoring of both conditions and effectiveness. Chadwick et.al. [1,2] have demonstrated that process conditions affect gas emissions. Apart from aeration, the parameters of the composting process with the highest impact on greenhouse gas emissions are mechanical agitation, moisture content, and temperature [3,4]. Other authors also point to the composition of the input material and its porosity [5]. On the other hand, some researchers are still concerned about this solution to waste treatment [6]. They show problems associated with the use of stabilized OFMSW and EU objectives set out in avoidance of biodegradable waste landfilling. This requires determination of its impact on the environment similar to the case of gas emissions.

Research into greenhouse gas emissions from composting has concerned mainly carbon dioxide ($CO_2$), methane ($CH_4$), and nitrous oxide ($N_2O$) [7], or has focused only on $CH_4$ [8] or $CO_2$ emissions [9]. Elimination of biodegradable material before placement in MSW landfills affects the minimization of the emission of landfill gases (LG), such as methane ($CH_4$) and carbon dioxide ($CO_2$), and their influence on global climate change [10]. A gas that is considered as not directly related to AB, is carbon monoxide (CO). CO is an odorless, colorless, and tasteless gas, exhibiting high toxicity to living organisms [11,12]. CO is an air pollutant commonly formed during natural and anthropogenic processes involving incomplete combustion and photochemical reactions. Much less is known about biological CO formation during decomposition of the organic fraction. Some authors show the role of organic and water content and temperature in CO production [13,14]. CO formation occurs due to thermochemical (abiotic) and biochemical (biotic) degradation. CO is formed as a result of the biological decomposition of the OF together with other gases such as $CO_2$, $CH_4$, $H_2$, nitrogen compounds, volatile organic compounds (VOCs), and $H_2S$ [15]. CO formation has been observed during composting of green waste [16], green waste with manure [17], organic waste [18,19], and municipal waste [20,21]. The discovery of CO formation during composting was unexpected. Thus, studying CO formation during these common, natural processes is needed for developing advanced biowaste treatment technologies involving sustainable carbon cycling.

This research aimed to obtain data of the fluctuation of CO content as well as that of $O_2$, $CO_2$, and $CH_4$ in process gas within the waste pile, during the AB of the OFMSW. The technical scale biostabilization process was carried out with OFMSW in six reactors under different technological regimes. The CO formation, $O_2$ and $CO_2$ concentrations, and temperature inside the reactor were measured at weekly intervals.

The dataset (consisting of the raw data in the Excel spreadsheet Supplementary materials.xls in the Supplementary Materials) show that the CO concentration was highly variable during the process, and the presence of the gas was recorded throughout the entire process (Table S5). The lowest CO concentration was recorded in reactors A1 and A2 and in reactors C1 and C2 (Table S1). Conditions for the highest carbon monoxide generation were in reactor B1, in which CO concentrations nearly doubled when the process was shortened by three weeks (Table S5). Such a high average concentration in the reactor B2 arose from high concentrations observed at the beginning of the process, because towards the end, results were similar to those observed in other reactors. The highest concentration of CO was observed between the second and fifth weeks of the test. The reactor B2 was the source of the highest CO production and average highest temperature. In this reactor, aeration was moderately intensive, which has a huge impact on CO concentrations. Thus, the main CO production occurred more or less at the same time as the highest temperatures in the reactors. This study suggests that the

highest CO production occurs at the highest temperature, which is why the authors suggest that CO production has mostly thermochemical foundations.

## 2. Data Description

The results obtained during the technical-scale experiments are presented in the form of five tables in the Supplementary Materials as an Excel spreadsheet file Supplementary materials.xlsx. There is also a readme sheet with the list of tables.

Table S1. Basic technological parameters of the reactors (Sheet 1: S1)

Table S2. The dimensions of tested reactors (Sheet 2: S2)

Table S3. Characteristics of organic fraction of municipal solid waste (OFMSW) properties (Sheet: S3)

Table S4. Changes in the morphological composition of OFMSW: samples were collected during the first and last day of the aerobic biostabilization (AB) process and represent the entire reactor (Sheet 4: S4)

Table S5. The concentration of measured gases and temperatures during the AB of OFMSW (Sheet 5: S5)

## 3. Methods

### 3.1. Waste Characteristics

The OFMSW used in the experiments was municipal waste with the <80 mm fraction mechanically separated. MSW originated from Warsaw, Poland. For waste characterization, six samples before the stabilization (samples characterizing the initial properties of the waste) and six samples after the stabilization (samples characterizing the final properties of the waste) were taken, which gave twelve samples of waste in total. Each sample was collected according to the following procedure: from four places along the length of the reactor, ~10 samples were taken. All collected samples from one reactor at the same collection moment were mixed and using the quartering method they were assigned to a representative sample. Samples were tested for pH reaction in accordance with Polish standard PN-EN-15011-3:2001, the moisture content in accordance with Polish standard PN-EN 14346:2011, loss on ignition (LOI) in accordance with PN-EN 15169:2011, the total organic carbon in accordance with Polish standard PN-EN 15936:2013 (Table S3), and morphological composition in accordance with Polish standard PN-93/Z-15006 (Table S4).

### 3.2. Experiment Configuration

The measurements were carried out between 24 April and 9 September 2015 at an industrial-scale waste treatment plant of the Municipal Cleaning Company in Warsaw, Poland. Six full cycles of AB of the OFMSW in bioreactors were provided. For each bioreactor, waste was placed along the aeration channels by a loader (Figure 1), and the waste pile was covered by a semipermeable membrane. The membrane was used to eliminate emission of odor and to prevent inflow of rainwater into the waste reactor yet allowed penetration of purified air and vapor. Finally, a length of firehose filled with sand was placed along the edge of the reactor to protect against the membrane blowing away (Figure 1). After that, the air blower was turned on, and the process started. Waste samples were taken at the beginning of the process (initial sample) and after six or nine weeks of the stabilization process (final sample). The distribution of temperature and gases inside the reactor was measured at weekly intervals. The scheme of the experiment is shown in Figure 2.

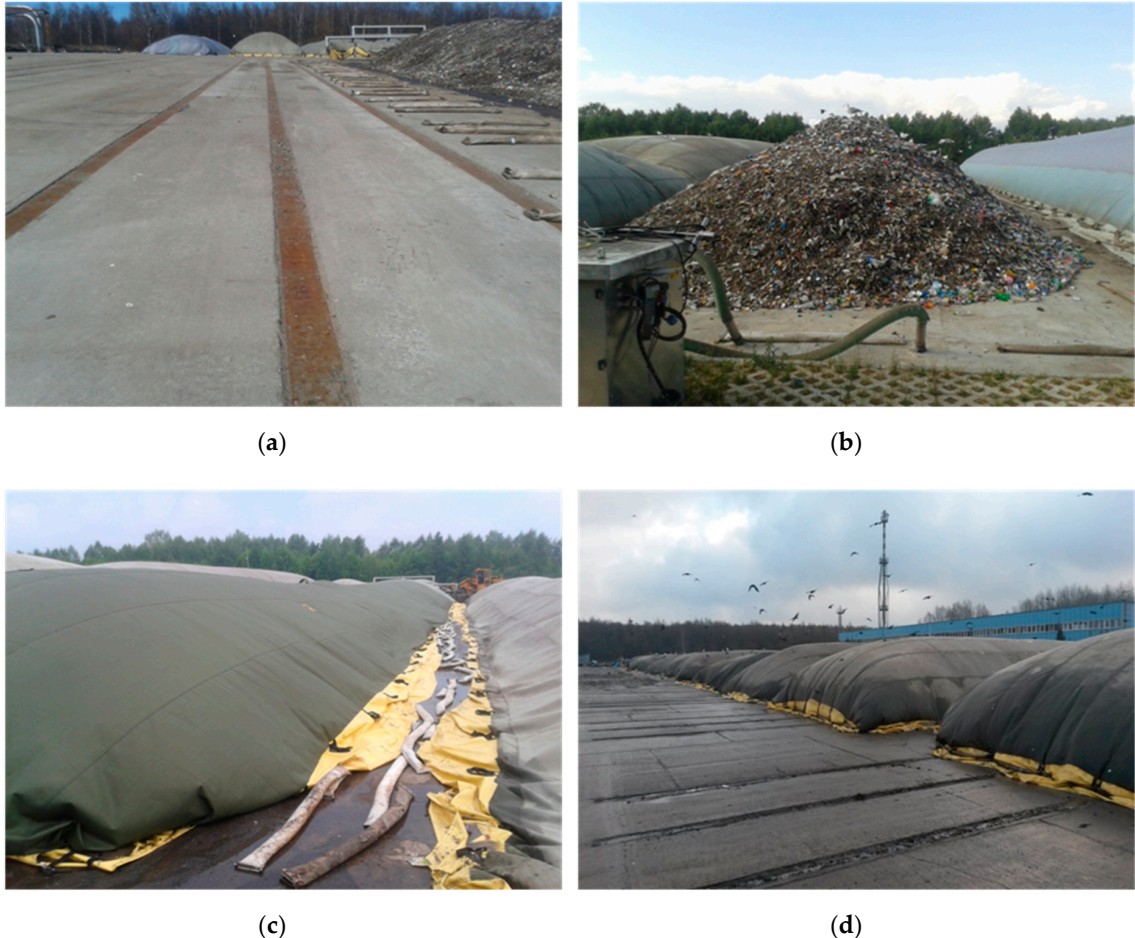

(**a**)　　　　　　　　　　　　　　　　　　　　　　(**b**)

(**c**)　　　　　　　　　　　　　　　　　　　　　　(**d**)

**Figure 1.** The biostabilization of OFMSW (**a**) base of the reactors with aeration channels, (**b**) OFMSW placed in the reactor before covering, (**c**) waste covered with a semi-permeable membrane, (**d**) reactors during aeration.

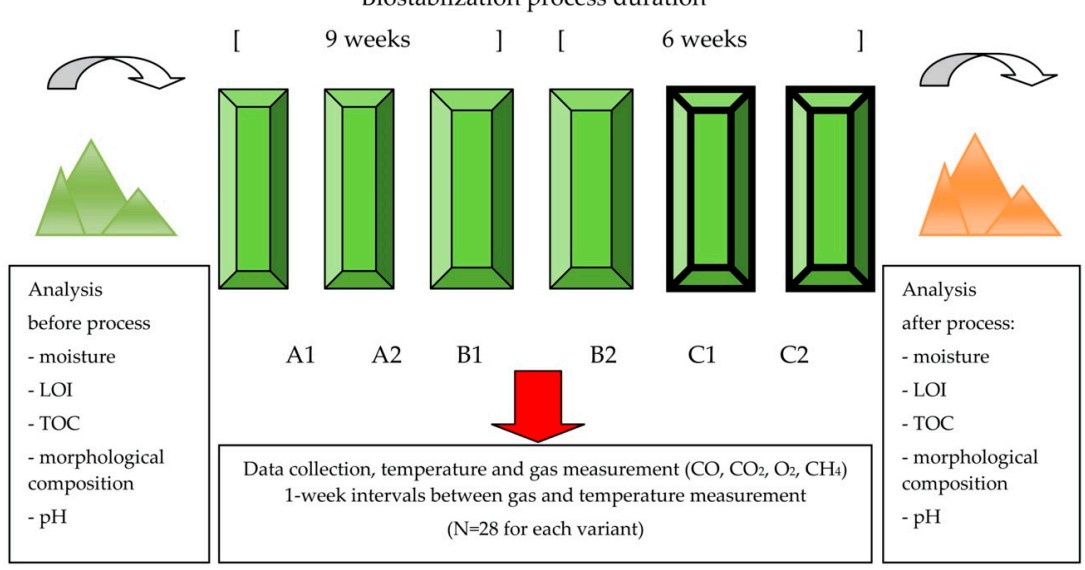

**Figure 2.** The scheme of experiment setup.

During the AB process, the formation of gases was dependent on process duration (42–69 days), waste mass (391.02–702.38 Mg), the intensity of waste aeration (4.4–10.7 $m^3 \cdot Mg^{-1} \cdot h^{-1}$), reactor design (membrane-covered reactor or membrane-covered reactor with sidewalls) and thermal conditions in the reactor (20.2–77.0 °C).

The first two reactors (A1 and A2) were loaded with ca 400 Mg of waste and the B1, B2, C1, C2 reactors with 600 to 700 Mg of waste (Table S1).

The influence of reactor design and operation on emissions was also tested. Reactors A1, A2, B1, and B2 were each covered with a semipermeable membrane attached to the ground, while in reactors C1 and C2 the waste material was loaded between sidewalls (height 1.5 m) extending the length of the pile, over which the membrane was stretched.

Waste retention times were varied between reactors. The full cycle of waste stabilization in the reactors lasted nine weeks in reactors A1, A2, and B1 and six weeks in B2, C1, and C2. The process was also carried out with variable total aeration ranging from 3,365 to 12,744 $m^3 \cdot Mg^{-1}$ of waste (Table S1).

The analyzed process parameters, such as physicochemical properties of the material, aeration intensity, thermal conditions, waste mass in the reactor, and reactor design were referenced to changes in the concentration of CO and other greenhouse gases (Table S1). The full configuration of the experiment is presented in Figure 2.

### 3.3. Measurements of Gas Concentrations in the Reactors

Gas concentrations were measured with a steel probe with holes at the end. Inside the probe, plastic wire was placed to help transport gas to the electrochemical analyzer, Kigaz 300 by Kimo (Kimo Instruments, Chevry-Cossigny, France), as well as a thermocouple, used to measure the temperature inside the reactors. A Kigaz 300 analyzer is commonly used as a portable electrochemical device for the determination of composition exhaust (flue) gases from thermal processes. $O_2$ and $CO_2$ volumetric contents in piles were measured as 0%–21% (±0.1%), but CO 0–2000 ppm and $CH_4$ 0–10,000 ppm contents were measured in volumetric ppm ±1 ppm. Although the measuring range of CO can be up to 2000 ppm, to ensure the safety of measurements and increase the equipment life, the measurements were stopped when the concentration exceeded 1000 ppm. In the dataset, these values were marked as >1000 ppm. The authors recommend deleting this data, for statistical calculations, due to unknown real values. The procedure of calibration of the Kigaz 300 analyzer was as described in our previous study [18]. Each time the analyzer was started, it performed a 2 min autocalibration. After placing the probe in the measuring location, the gas concentration values stabilized, typically within ~5 min. After each measurement, the probe was removed from the compost for ~1 min, allowing the measured gas values to return to ambient air.

Measurements were taken along the length of the reactor at the following distance points: 2.5, 17.5, 32.5, and 47.5 m from the fan (Figure 3). At each of these points, three measurements were taken at various heights (H) within the reactor, which differed depending on the size of the reactor. Details are presented in Table S2. The measurements were taken separately for the right and left sides of the reactor. The left side of the reactor was the side on the left looking from the fan. In addition, on one side of the reactor at each distance point at the middle height, the so-called deep measurement was taken to illustrate the concentration of gases inside the reactor (Figure 3). During the test of the reactors at the 28 selected points, a total of 1,080 gas and temperature samples were taken. Data obtained at the technical scale carries the risk of errors (for example equipment failures during field study), but this was reduced by a large number of repetitions each time (28 measurements in each pile). All data that provide an unexpected condition were eliminated from the dataset in advance of future statistical analysis. As the experiment was done at a technical scale under normal operation (B1) with some technological modifications (A1, A2, B2, C1, C2), there is no variant zero (control), but all data should be compared to pile B1, which describes the standard AB procedure implemented in that installation. Detailed information about the methodology of process and temperature measurements is provided in our previous study [18].

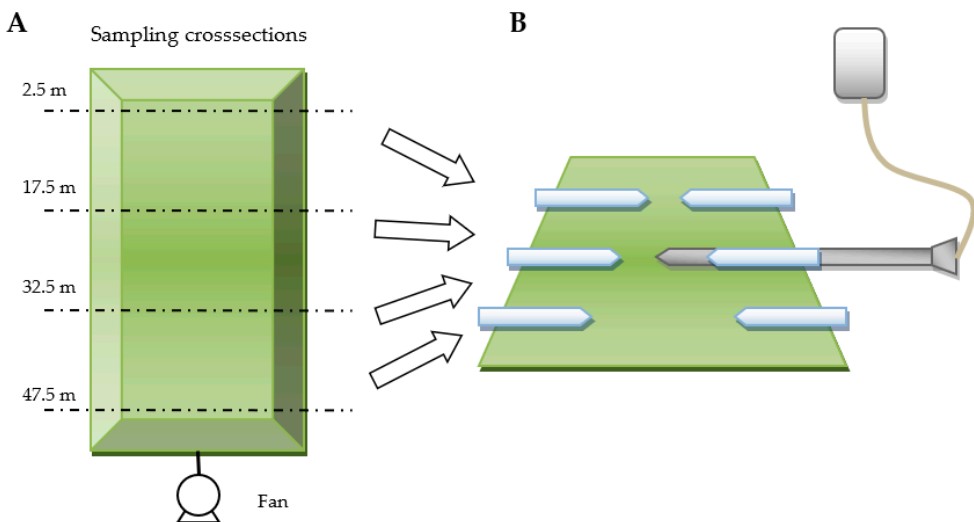

**Figure 3.** The scheme of gas and temperature sampling points in the reactors. Sampling cross-sections (**A**) Position of gas and temperature sampling points (shallow: blue and deep: grey) (**B**).

## 4. User Notes

The presented dataset may be useful to study the spatial and temporal distribution of temperature and the CO, $O_2$, $CO_2$, and $CH_4$ content during OFMSW aerobic biostabilization. It may also be useful for the study of the influence of technological parameters on temperature, CO content, and $O_2$, $CO_2$, and $CH_4$ fluctuations during OFMSW aerobic biostabilization by different types of regression analyses. Authors recommend deleting all data exceeding the measuring scale before statistical analysis.

**Supplementary Materials:** The following are available online at http://www.mdpi.com/2306-5729/5/2/40/s1: Excel file Supplementary materials.xlsx contains six sheets as follows: readme sheet with a list of tables (Sheet 0); Table S1. Basic technological parameters of the reactors (Sheet 1); Table S2. The dimensions of the tested reactors (Sheet 2); Table. S3. Characteristics of organic fraction of municipal solid waste (OFMSW) properties (Sheet 3); Table S4. Changes in the morphological composition of OFMSW: samples were collected during the first and last day of the AB process and represent the entire reactor (Sheet 4); Table S5. The concentration of measured gases and temperature during the aerobic biostabilization (AB) of OFMSW (Sheet 5).

**Author Contributions:** Conceptualization, A.B.; methodology, A.B.; validation, A.B., J.B., P.F.R.; formal analysis, S.S.-D., J.R., A.B., P.B.; investigation, S.S.-D., J.R., A.B, P.B., M.D.; resources, A.B.; writing—original draft preparation, S.S.-D., A.B.; writing—review and editing, S.S.-D., A.B., P.F.R.; visualization, S.S.-D.; supervision, A.B., J.B.; project administration, A.B.; funding acquisition, A.B. All authors have read and agreed to the published version of the manuscript.

**Funding:** This research was funded by the Municipal Cleaning Company in Warsaw, Poland, "The investigation on the intensity of the aerobic biological processes occurring in the prisms for biostabilization of the municipal solid waste undersize fraction".

**Conflicts of Interest:** The authors declare no conflict of interest.

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
