# Peer review of "The Fluctuation of Process Gasses Especially of Carbon Monoxide during Aerobic Biostabilization of an Organic Fraction of Municipal Solid Waste under Different Technological Regimes"

_data_

Round 1
Reviewer 1 Report
In this dataset, it is summarized the temperature and the content of process gases (including rarely reported carbon monoxide, CO) generated inside full-scale aerobic biodegradation of OFMSW reactor.
The data are original and the source is well defined.
The methods and materials are well defined. The data have been clearly archived to make them available to other authors.
In order to improve this Data Descriptor, however, it is considered necessary:
- adequately describe quality control measures and/or possible sources of error;
- verify some typing errors:
Line 124: ... to eliminate emission of odoe and to prevent
Line 126: ...a lengtg offirehosefilled
Author Response
Dear Reviewer,
Thank you very much for all your comments. Hereafter you can find our replies to your suggestions/comments when it was possible to apply it, and our answer in the attachment.
All changes made within the manuscript according to your comments are highlighted in red color.

Reviewer 2 Report
Dear Authors, the paper "The fluctuation of process gasses especially of carbon monoxide during aerobic biostabilization of an organic fraction of municipal solid waste under different technological regimes" is an intersting data descriptor. To improve the paper I would suggest to show the main challenges in the introduction and then show how these were met in the conclusion part.
The presented literature is rather poor and quite not up-to-date, I believe the Authors show consider to expand the reference list in order to highight the novelty and originality of the work, especially for a broad reader.
Abstract is a good representantion of the papers text. However if you discuss the extremal values of the parameters, the range of the experiment must be mentioned.
Consider to change the subtitle for first chapter, it is in fact introduction.
The data set is prepared carefully, the data are original, the methodologies of data collection are described with sufficient details and adhere to relevant disciplinary standards. However, the dataset is based on single run, control is missing, as not even SD is provided. Please refer.
The datasets technical soundness is suficient, but appropriate quality control measures were not employed and described. The possible sources of error and noise are not described.
The description of the dataset is supported by the results. The user set explaind possible application sufficiently.
Author Response

(The authors gave the same response as above.)
